# Experimental Investigation of Friction Stir Welding on 6061-T6 Aluminum Alloy using Taguchi-Based GRA

**Assefa Asmare** [1]**, Raheem Al-Sabur** [2] 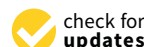 **and Eyob Messele** [1,*]

[1]  Faculty of Mechanical and Industrial Engineering, Bahir Dar Institute of Technology, Bahir Dar University, Bahir Dar 6000, Ethiopia; assefaa@bdu.edu.et

[2]  College of Engineering, University of Basrah, Basrah 6100, Iraq; Raheem.musawel@uobasrah.edu.iq

*   Correspondence: Eyob.Messele@bdu.edu.et or Eyobsmart27@gmail.com; Tel.: +251-918-161601

**Abstract:** The use of aluminum alloys, nowadays, is swiftly growing from the prerequisite of producing higher strength to weight ratio. Lightweight components are crucial interest in most manufacturing sectors, especially in transportation, aviation, maritime, automotive, and others. Traditional available joining methods have an adverse effect on joining these lightweight engineering materials, increasing needs for new environmentally friendly joining methods. Hence, friction stir welding (FSW) is introduced. Friction stir welding is a relatively new welding process that can produce high-quality weld joints with a lightweight and low joining cost with no waste. This paper endeavors to deals with optimizing process parameters for quality criteria on tensile and hardness strengths. Samples were taken from a 5 mm 6061-T6 aluminum alloy sheet with butt joint configuration. Controlled process parameters tool profile, rotational speed and transverse speed were utilized. The process parameters are optimized making use of the combination of Grey relation analysis method and L$_9$ orthogonal array. Mechanical properties of the weld joints are examined through tensile, hardness, and liquid penetrant tests at room temperature. From this research, rotational speed and traverse speed become significant parameters at a 99% confidence interval, and the joint efficiency reached 91.3%.

**Keywords:** friction stir welding; AA6061; grey relation; orthogonal array; Anova

## 1. Introduction

Nowadays in the transportation industry, the stipulate for lightweight, and higher strength structures are augmented swiftly for diminutions of fuel consumption and enhancement of payload capability [1–3]. One of the ways to diminish the weight of parts is the utilization of an advanced joining process. At this contemporary epoch, one of the latest and advanced joining processes that offers numerous advantages, especially the joining of lightweight materials, is known as friction stir welding [4,5]. Friction stir welding (FSW) is one of a solid-state joining process that is comparatively a new welding technique invented by W. Thomas and E. Nicholas at The Welding Institute (TWI) of Cambridge in the United Kingdom in 1991. The process is suitable for welding of ferrous and non-ferrous materials of the same kind or dissimilar, especially recommended for soft materials namely aluminum, copper, nickel, titanium, and others [6–10]. The utilization of lightweight materials is a crucial point of interest [11] and most of the constructional and structural materials are increasingly replaced by non-ferrous materials, such as aluminium alloys, and a combination of both (steel and aluminum) that aids to diminish the cost and the weight problems in many industrial applications [1]. One of the lightweight materials under the metallic category is aluminum AA 6061-T6 [3]. In recent years, AA 6061-T6 is the most usual and commonly used in the aviation, maritime, and automotive industries, due to its weight saving, higher strength, and machinability features. AA 6061-T6 material

is particularly appropriate for the welding of high strength alloys which are extensively used in the vehicle and aircraft industry [12]. In addition to this, most of the transportation industries, i.e., depicted on the above utilize friction stir welding for minimizing the manufacturing period from 23 to 6 days, improved dimensional accuracy, increased the joint strength by a considerable amount when compared to fusion-welding, and reduced weight and electrical consumption [13]. Consequently, AA 6061-T6 and friction stir welding is an important material, and joining processes for transportation industries due to its outcomes. However, in this joining technique, the process parameters play a vital role in affecting the mechanical plus the metallurgical properties of the weldment [14–17]. The higher tool rotational and the lower traverse speeds are the source of producing adequate heat for joining the base metal and those are the highest statistical influence on hardness, tensile strength, and peak temperature relative to other parameters [18–20]. On the other hand, the tool pin profile is governing the material movement and significantly influence the flow of plasticized material. It has a strong effect on material flow in the weld nugget zone. The shape of the tool pin profiles affects the mechanical properties of the weld joint [21,22]. Generally, improper selection of FSW process parameters results in defect formation, which in turn deteriorates the mechanical properties of the joint. This may happen due to the amount of heat generation that affects the formation of defects in FSW [23]. The most known defects are tunnel, flash, kissing bond, void/wormhole, cavity/groove, and crack defects [24]. Those defects can be controlled by optimizing the process parameters and utilizing suitable tools. However, welding of AA 6061-T6 by FSW needs appropriate process parameters. This topmost problem is the relationship between welding parameters and mechanical properties; it needs a systematic study. Hence, this paper deals with the finding of optimum process parameters for enhancing the quality criteria on hardness and tensile strength of the target material. Ishikawa or cause and effect diagram developed in Figure 1 sorts out the possible root causes of FSW defects and identifies suitable and control welding process parameters that impart good welding quality for the target material. Due to multi quality criteria, the hybrid Grey Relational Analysis (GRA) and Taguchi method executed to optimize the process parameters validates the results.

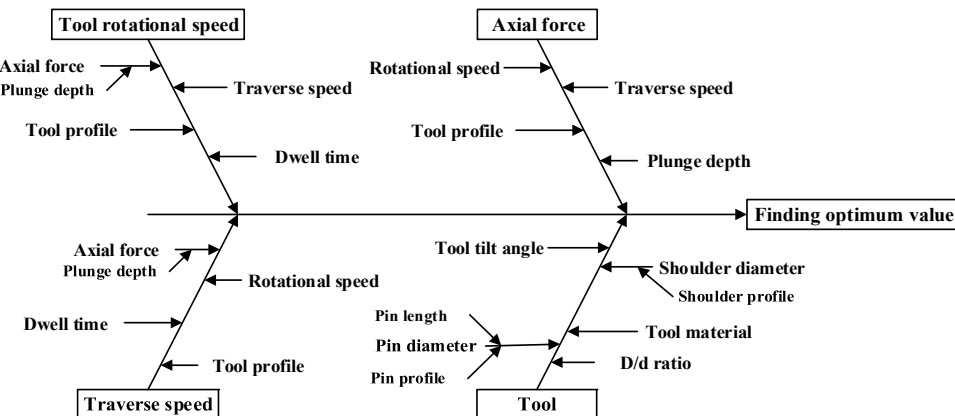

**Figure 1.** Cause and effect diagram of parameters influencing on FSW joint quality.

## 2. Materials and Methods

### 2.1. Sample Preparation

The material used in this study was 6061-T6 aluminum alloy with a butt joint configuration. The chemical, mechanical, and thermo-physical properties of this material are depicted in Tables 1–3, respectively. The welding sheet was cut off parallel into the rolling direction with a dimension of 101.6 mm × 20 mm × 5 mm using a hand hacksaw to minimize the residual stresses that will occur during the cutting operations. In addition, the tensile strength test samples were prepared according

to ASTM E8-04 [25] standards as shown in Figure 2 by making use of a metalcraft VMBS 1610 band saw machine.

**Table 1.** Chemical composition of AA 6061 material.

| Material % | Mg | Si | Fe | Cr | Cu | AL |
|---|---|---|---|---|---|---|
| AA 6061 | 0.92 | 0.6 | 0.33 | 0.18 | 0.25 | 97.72 |

**Table 2.** Mechanical property of AA 6061 material.

| Material | Yield Strength (MPa) | Ultimate Tensile Strength (MPa) | Hardness (HRA) |
|---|---|---|---|
| AA6061 | 276 | 310 | 40 |

**Table 3.** Thermo-physical properties of 6061 aluminum alloy material [26,27].

| Density (g/cm$^3$) | Melting Point (°C) | Thermal Conductivity (W/m-k) | Specific Heat (J/Kg-°C) |
|---|---|---|---|
| 2.7 | 652 | 167 | 0.896 |

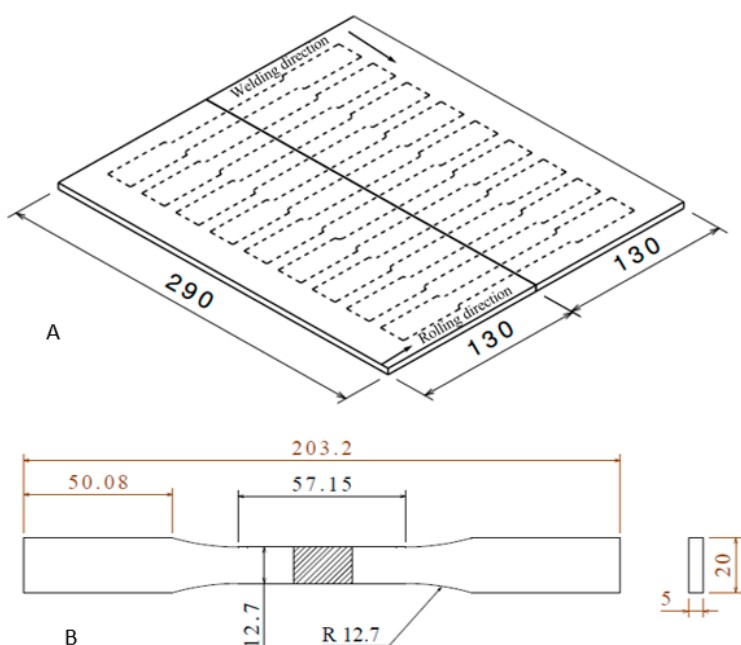

**Figure 2.** Pattern of welding with respect to rolling direction and removal of tensile specimen: (**A**) Dimension of the flat tensile specimen, (**B**) welded specimen according to (ASTM E8-04).

*2.2. Experimental Setup*

The experimental setup depicted in Figure 3 was carried out on a high-precision XHS7145 vertical CNC machine (Shanghai Bairuo Testing Instrument Co., Ltd., Shanghai, China) with a maximum speed of 8000 rpm. The welding tool was used for the experimentation made from H13 tool steel with different pin profiles. The mechanical property and chemical composition of H13 tool are listed respectively in Tables 4 and 5. The tool pin length geometrical dimension, represented in Figure 4, was prepared at 0.3 mm less than the base metal thickness for minimizing the tool wear during the welding [28]. The tensile strengths of the weld joint samples are depicted in Figure 5 and were measured by the Bairoe computer controlled electro-hydraulic universal testing machine (Shandong Lunan machine Tool (Group) Co.,Ltd., Shandong, China) of model HUT-600. Besides, the hardness of the joint was measured by the Rockwell hardness-testing machine in scale A. The transient heating that occurs during the welding process was measured by K type thermocouples at a center symmetry point on the

advancing and retreating side of the specimen. Digital data logger with the integration of compression type load-cell and data transmitter controlled the axial force of the tool. The parameters identified for this study are tool rotation speed, traverse speed, and tool pin profile. The selected process parameters and their levels are shown in Table 6. Moreover, the quality of FSW was governed by utilizing a firmly secured fixture [29]. Hence, the workpiece fixture was designed to avoid any unwanted free vibrations.

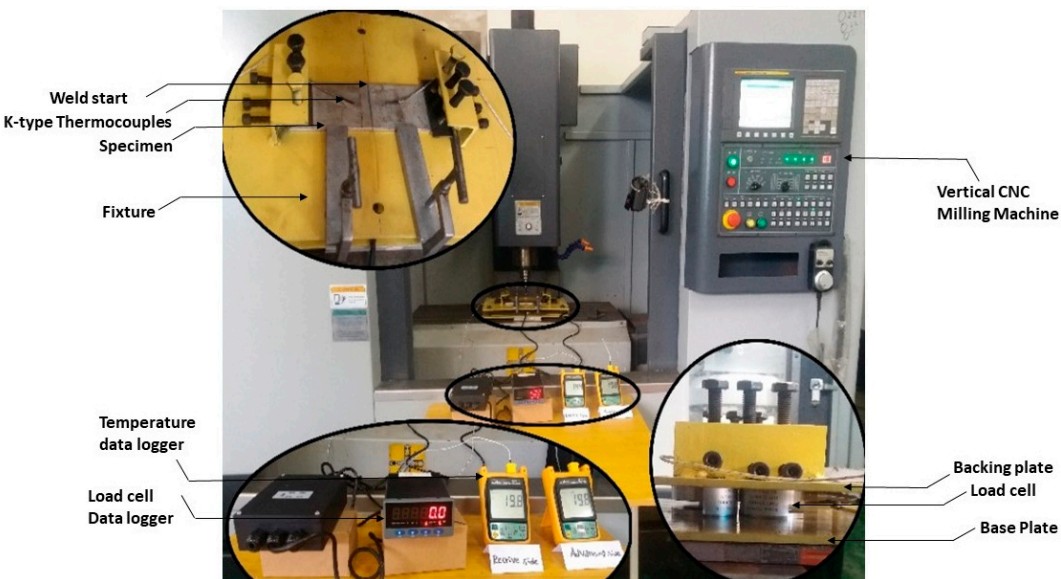

**Figure 3.** Experimental setup used for FSW connected with the load cell and thermocouples.

**Table 4.** Mechanical properties of H13 tool steel [30,31].

| No | Property at 20 $^\circ$C | Mechanical Strength |
|----|--------------------------|---------------------|
| 1 | Ultimate tensile strength | 1200–1590 MPa |
| 2 | Hardness | 53 HRA |
| 3 | Yield strength | 1000–1380 MPa |
| 4 | Modulus of elasticity | 215,116,427,448 N/m$^2$ |
| 5 | Reduction of area | 50% |
| 6 | Poisons ratio | 0.27–0.3 |

**Table 5.** Chemical composition of H13 tool steel [32].

| Material | %C | %Si | %Cr | %Mo | %V |
|----------|-----|------|------|-----|-----|
| H13 | 0.4 | 1.00 | 5.30 | 1.4 | 1.0 |

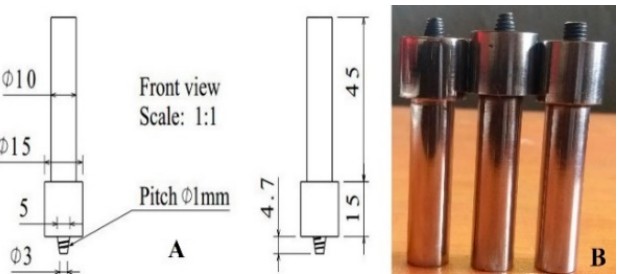

**Figure 4.** (**A**) Geometric dimensions of the tool, (**B**) From left to right tri-flute threaded, taper threaded and cylindrical threaded tools.

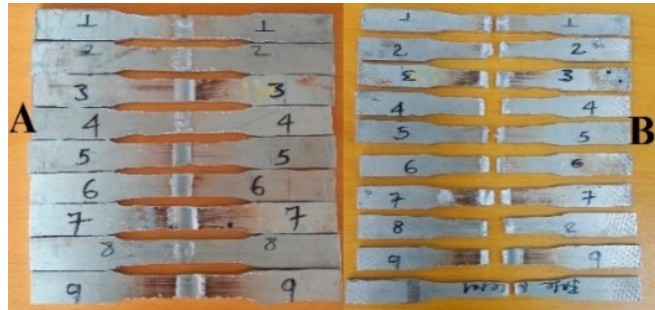

**Figure 5.** Extraction of tensile specimens: (**A**) Welded specimens before the tensile test; (**B**) welded specimens after the tensile test.

**Table 6.** Control process parameters and their levels.

| Parameters | Level | | |
| --- | --- | --- | --- |
| | 1 | 2 | 3 |
| Tool profile | Cylindrical | Taper | Tri-flute |
| Rotational Speed, rpm | 900 | 1200 | 1400 |
| Traverse speed, mm/min | 37.5 | 42.5 | 47.5 |

## 3. Results and Discussions

The experiments were conducted to study the effect of process parameters over the output response characteristics of hardness and tensile strengths and are summarized as shown in Table 7.

**Table 7.** Taguchi $L_9$ orthogonal array parameter setting and experimental results of hardness and tensile strength.

| No. | Tool Profile (Type) | Rotational Speed (rpm) | Traverse Speed (mm/min) | UTS (MPa) | HR (HRA) |
| --- | --- | --- | --- | --- | --- |
| 1 | Cylindrical | 900 | 37.5 | 253 | 59.44 |
| 2 | Cylindrical | 1200 | 42.5 | 263 | 65.70 |
| 3 | Cylindrical | 1400 | 47.5 | 272 | 67.70 |
| 4 | Taper | 900 | 42.5 | 231 | 56.00 |
| 5 | Taper | 1200 | 47.5 | 254 | 63.66 |
| 6 | Taper | 1400 | 37.5 | 283 | 71.60 |
| 7 | Tri-flute | 900 | 47.5 | 217 | 54.23 |
| 8 | Tri-flute | 1200 | 37.5 | 276 | 69.10 |
| 9 | Tri-flute | 1400 | 42.5 | 281 | 69.80 |

*3.1. Tensile Strength*

Tensile strength is one of the responses that was measured triple times at room temperatures for similar FS welded of Al-alloys (6061). The effect of rotational speed on the tensile strength has been shown in Figure 6 and the highest tensile strength of 283 MPa was observed from a tapper pin profile tool at a rotational speed of 1400 rpm, and traverse speed of 37.5 mm/min and its joint efficiency was reached about 91.3%. Similarly, the lowest tensile strength of 217 MPa was observed at 900 rpm, traverse speed of 47.5 mm/min, and tri-flute threaded tool pin profile. The result shows that hardness and tensile strength are directly proportional to the rotational speed and inversely proportional to the traverse speed of the tool for this reason: that the lower traverse speed and higher rotational speed produce adequate heat for joining the base metal.

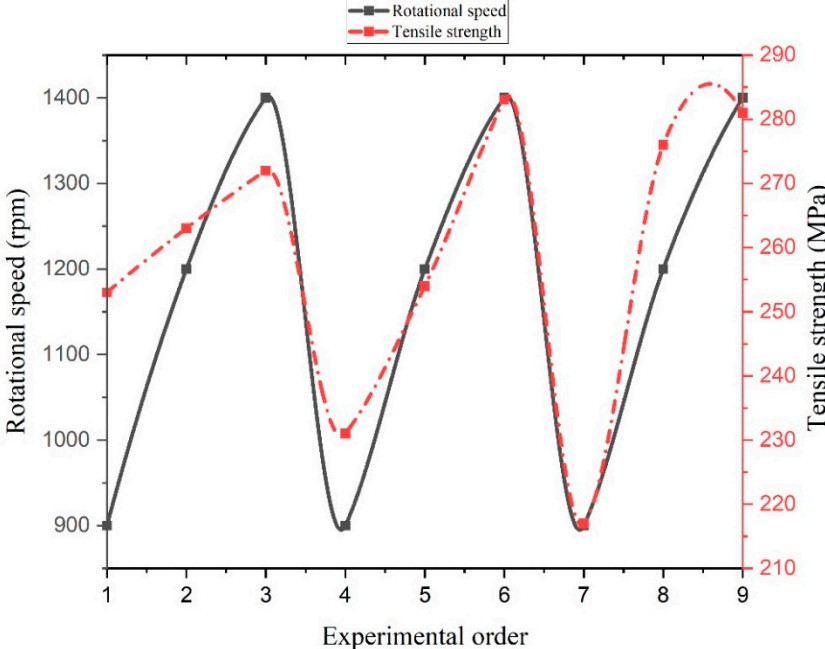

**Figure 6.** Effects of rotational speed on the tensile strength property of AA 6061.

*3.2. Hardness*

The hardness of the joint was measured three times at the nugget zone. The higher hardness value of 71.6 HR was obtained at the pick point of the curve in Figure 7 at a parameter setting of a rotational speed of 1400 rpm, traverse speed of 37.5 mm/min, and taper threaded tool pin profile. Correspondingly, the minimum hardness value of 54.23 HR was recorded at a rotational speed of 900 rpm, traverse speed of 47.5 mm/min, and tri-flute threaded tool pin profile. The maximum rotational speed with a combination of a taper threaded tool pin imparts the highest hardness strength.

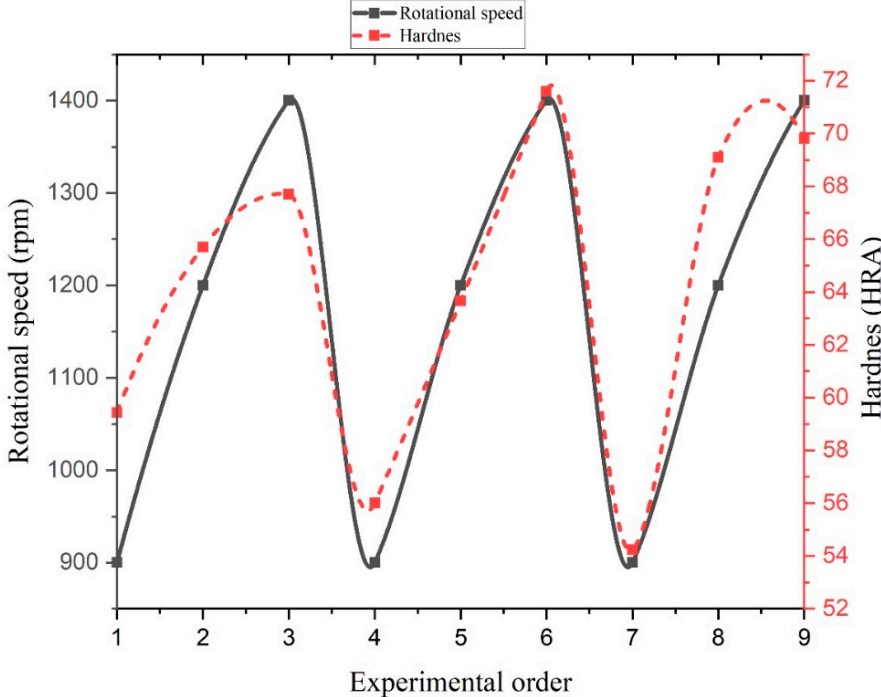

**Figure 7.** Effects of rotational speed on the hardness property of AA 6061.

### 3.3. Liquid Penetrant Test

As shown in Figure 8, the test was performed on all 9 experiments along the joint line of the weld. The result shows that Experiment 1, 4, and 8 have a present visible discontinuity along the weld joint. Those experiments, comparatively to the other joints, are more defective. The remaining experiments have a defect-free joint. Surface cracks exist onall of the weld joints at the start and endpoints of the joint. Therefore, the higher rotational speed due to its high heat input and friction delivers defect-free joints. On the other hand, the lower rotational speed has some defects along the welded joints.

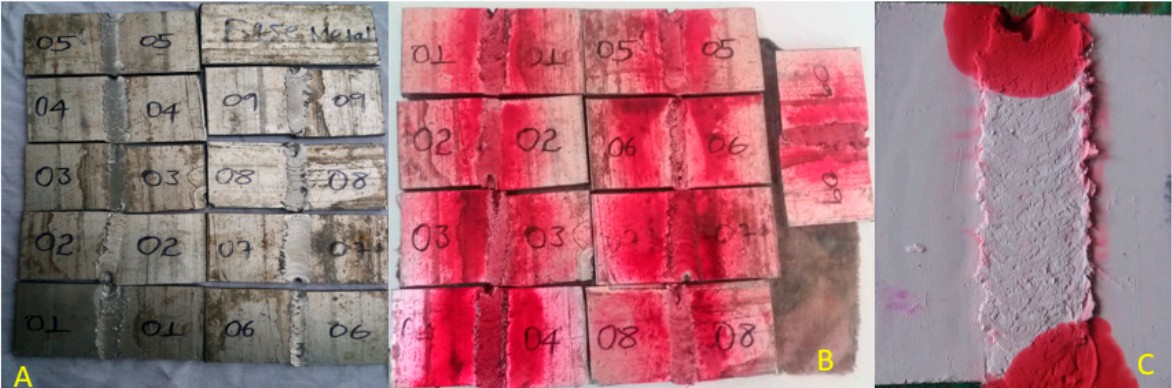

**Figure 8.** Liquid penetrant test: (**A**) Welded specimens before the liquid penetrant test; (**B**) welded specimens after applying penetrants on the faying surfaces; (**C**) welded specimens after applying developer.

### 3.4. Effect of Welding Parameters on the Temperature Profiles

Temperatures on the advancing and retreating sides are shown in Table 8, below.

**Table 8.** Correlation of welding temperature on the selected parameters.

| No. | Traverse Speed (Mm/min) | Rotational Speed (RPM) | Max. Temp. on Advancing Side (°C) | Max. Temp. on Retreating Side (°C) |
|---|---|---|---|---|
| 1 | 37.5 | 900 | 342 | 328 |
| 2 | 42.5 | 1200 | 388 | 371 |
| 3 | 47.5 | 1400 | 396 | 380 |
| 4 | 37.5 | 900 | 301 | 283 |
| 5 | 42.5 | 1200 | 374 | 358 |
| 6 | 47.5 | 1400 | 416 | 402 |
| 7 | 37.5 | 1200 | 319 | 296 |
| 8 | 42.5 | 1400 | 404 | 393 |
| 9 | 47.5 | 900 | 412 | 399 |

Based on the results of this study, the temperature increased when rotational speed increased due to the severe plastic deformation caused by the high stirring process. At all experiments, the advancing side is affected by a higher temperature than the retreating side shown in Figure 9. Besides, the temperature is directly proportional to the rotational speed and inversely proportional to the traverse speed.

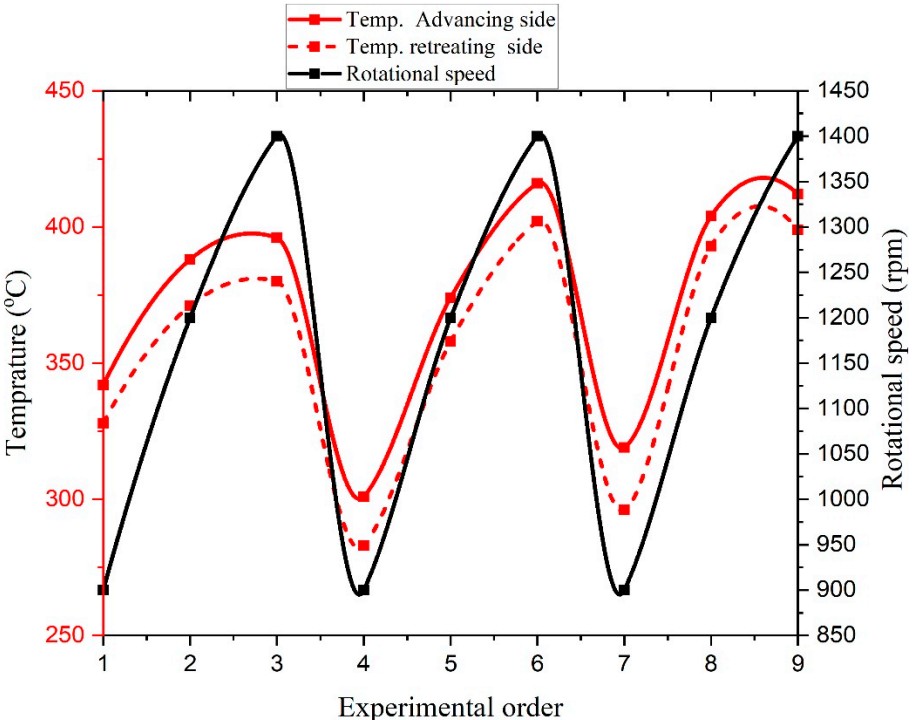

**Figure 9.** Correlation between rotational speed and welding temperature.

### 3.5. Welding Parameter Effects on the Joint Quality

In the friction stir welding welding process, a variety of process parameters affects the weld joint quality. FSW welding process parameters mainly include rotational speed of a tool, welding traverse speed and tool profile. Each experimental execution observations have discussed and summarized in Table 9.

Experiments 3 and 5 have completely defect-free joints and Experiments 1, 7, and 9 have some flash defects on the advancing sides; but, due to a high RPM and welding speed, they bestow maximum hardness and tensile strength. Therefore, 1200 and 1400 rpm impart defect-free joints for 6061 materials.

**Table 9.** Rotational speed and traverse speed effect on the response study.

| RPM | T.S | Strength Property | Welding Joint | Observation |
|-----|-----|-------------------|---------------|-------------|
| 900 | 37.5 | T.S = 248<br>HR = 49.44 | | Name of the defect: Flash<br>(i)   Location of the defect: A.S<br>(ii)  Reason for the defect: Lower welding speed |
| 1200 | 42.5 | T.S = 263<br>HR = 55.7 | | Name of the defect: Defect free<br>(i)   Location of the defect: Nil<br>(ii)  Reason for the defect: Adequate heat input |
| 1400 | 47.5 | T.S = 272<br>HR = 57.7 | | Name of the defect: Defect free<br>(i)   Location of the defect: Nil<br>(ii)  Reason for the defect: Adequate heat input |

**Table 9.** *Cont.*

| RPM | T.S | Strength Property | Welding Joint | Observation |
|---|---|---|---|---|
| 900 | 42.5 | T.S = 231<br>HR = 46 |  | Name of the defect: Tunnel<br>(i) Location of the defect: Stir zone<br>(ii) Reason for the defect: Too low RPM |
| 1200 | 47.5 | T.S = 254<br>HR = 53.66 |  | Name of the defect: Defect free<br>(i) Location of the defect: Nil<br>(ii) Reason for the defect: Adequate heat input |
| 1400 | 37.5 | T.S = 283<br>HR = 61.6 |  | Name of the defect: Tunnel<br>(i) Location of the defect: Stir zone, at the beginning<br>(ii) Reason for the defect: Inappropriate pin offset |
| 900 | 47.5 | T.S = 217<br>HR = 44.23 |  | Name of the defect: Flash<br>(i) Location of the defect: Advancing side<br>(ii) Reason for the defect: Very high RPM and lower welding speed. |
| 1200 | 37.5 | T.S = 275<br>HR = 58.1 |  | Name of the defect: Tunnel<br>(i) Location of the defect: Stir zone<br>(ii) Reason for the defect: Inappropriate pin offset |
| 1400 | 42.5 | T.S = 281<br>HR = 59.8 |  | Name of the defect: Flash<br>(i) Location of the defect: Advancing side<br>(ii) Reason for the defect: Very high RPM and lower welding speed. |

T.S—Tensile strength MPa and HR—Rockwell Hardness.

*3.6. Statistical Analysis*

3.6.1. Taguchi Method

Taguchi method is one of the quality enhancement approaches developed by Dr. Genechi Taguchi in Japan in 1940 [33]. The technique is simple, capable, and a systematic quality improvement method that allows independent estimation of the response with a minimal number of trials [34,35]. This method involves two major tools: Orthogonal array (OA) and S/N ratio. Based on Latin Square [36], an orthogonal array is employed to reduce variance and optimize process parameters. On the other hand, the signal-to-noise-ratio is used to measure process robustness and to evaluate deviation from desired values based on the selected quality characteristics. The signal-to-noise-ratio quality characteristics are categorized into three main groups: larger is better, nominal is best, and smaller

is better [34–37]. In this research, used larger is better quality characteristics for both hardness and tensile responses. Larger is better criterion can be calculated using the below Equation [38].

$$\frac{S}{N}(\eta) = -10 \log 10 \frac{1}{n} \sum_{i=1}^{n} \frac{1}{y^2_{ijk}}$$ (1)

where *n* is the number of replications and $y_{ijk}$ is the response value of the *i*th performance characteristic in the *j*th experiment at the *k*th trial.

### 3.6.2. Grey Relation Analysis (GRA)

The Taguchi trial method is suitable to govern the optimal settings of process parameters for a solitary or mono-objective characteristic. Contingent upon two or more responses, Grey Relational Analysis method (GRA) being Taguchi-based is preferable [37]. GRA is one of multiple response optimization tools used to conduct a relational analysis of the uncertainty of a system model and solving sophisticated interconnection among multi-objective responses [39,40]. Optimizing parameters using the GRA method, seven flow steps are employed [41] as shown in Figure 10.

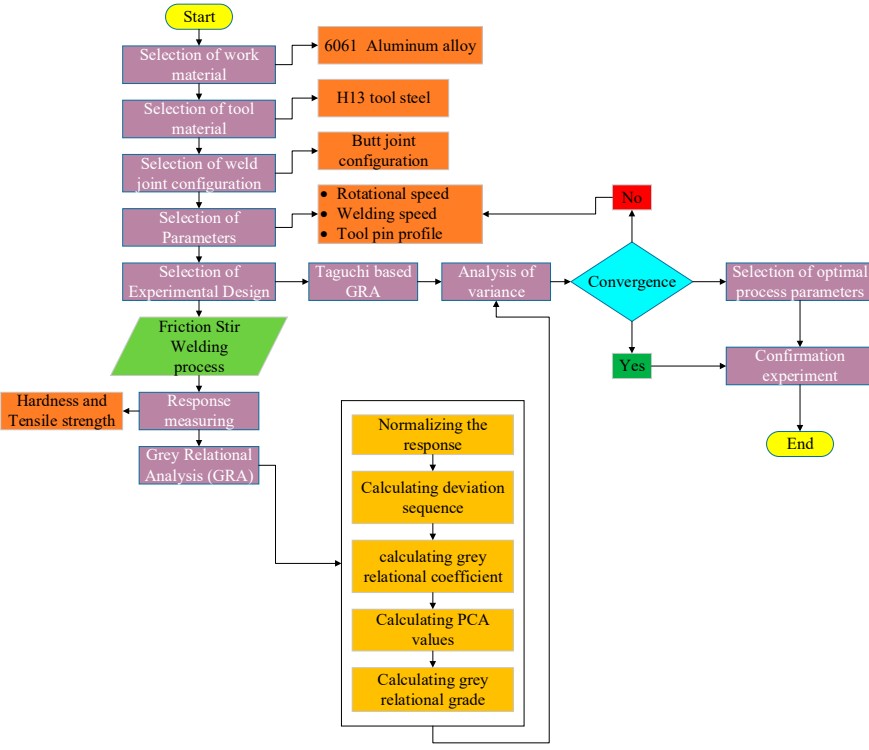

**Figure 10.** A research framework.

### 3.6.3. Data Normalization

Data normalization is the first step in the Grey relational analysis scenario. This step is carried out onces the signal-to-noise ration of the target quality creteria is obtained asshown in Table 10. This process is conveying the original sequence to a comparable sequence and the experimental results are normalized in the range between zero and one, due to different measurement units. This result is known as the Grey relational generation. This process is required when the sequence scatter range is too large, or when the direction of the target in the sequences are different [42–46]. If the response is to

be maximized, then larger is better, characteristics are intended for normalization to scale it into an acceptable range using Equation (2). The results of data normalization are depicted in Table 11 [38].

$$x_{i(k)=} \frac{x_i^0(k) - \min x_i^0(k)}{\max x_i^0(k) - \min x_i^0(k)} \tag{2}$$

**Table 10.** Multi-response experimental results with its signal to noise (S/N) ratio.

| No. | UTS (MPa) | HR (HRA) | $S/N_{UTM}$ | $S/N_{HR}$ |
|-----|-----------|----------|-------------|------------|
| 1 | 253 | 59.44 | 48.0624 | 35.4816 |
| 2 | 263 | 65.70 | 48.3991 | 36.3513 |
| 3 | 272 | 67.70 | 48.6914 | 36.6118 |
| 4 | 231 | 56.00 | 47.2722 | 34.9638 |
| 5 | 254 | 63.66 | 48.0967 | 36.0773 |
| 6 | 283 | 71.60 | 49.0357 | 37.0983 |
| 7 | 217 | 54.23 | 46.7292 | 34.6848 |
| 8 | 276 | 69.10 | 48.8182 | 36.7896 |
| 9 | 281 | 69.80 | 48.9741 | 36.8771 |

**Table 11.** Data normalization and deviation sequence.

| | Step 1: Data Normalized | | Step 2: Deviation Sequence | |
|-----|-----------|----------|-----------|----------|
| **No.** | **UTS (MPa)** | **HR (HRA)** | **UTS (MPa)** | **HR (HRA)** |
| 1 | 0.578 | 0.330 | 0.422 | 0.670 |
| 2 | 0.724 | 0.691 | 0.276 | 0.309 |
| 3 | 0.851 | 0.798 | 0.149 | 0.202 |
| 4 | 0.235 | 0.116 | 0.765 | 0.884 |
| 5 | 0.593 | 0.577 | 0.407 | 0.423 |
| 6 | 1.000 | 1.000 | 0.000 | 0.000 |
| 7 | 0.000 | 0.000 | 1.000 | 1.000 |
| 8 | 0.906 | 0.872 | 0.094 | 0.128 |
| 9 | 0.973 | 0.908 | 0.027 | 0.092 |

### 3.6.4. Deviation Sequences and Grey Relational Coefficients

The next step is finding a Grey relational coefficient(GRC), $\xi_i(k)$ from the normalized values using Equations (3) and (4). GRC used to explain the relationship between the reference sequence [44–47] and the comparability sequence. The GRC ($\xi$) is calculated to integrate the data achieved from Equations (3) and (4) and its calculated values are depicted in Table 15.

$$\Delta_{0i}(k) = \|x_{0^*}(k) - x_{i^*}(k)\| \tag{3}$$

$$\xi(x_{0^*}(k), x_{0^*}(k)) = \frac{\Delta_{min}(k) + \xi\Delta_{max}(k)}{\Delta_{0i}(k) + \xi\Delta_{max}(k)} \tag{4}$$

where $\Delta_{0i}(k)$ is the deviation sequence of the reference sequence $x_{0^*}(k)$ and comparability sequence $x_{i^*}(k)$, $\xi$ is the distinguishing coefficient that takes a value between 0 and 1, and the value of 0.5 is used based on the principal component analysis result. As given in Equation (4), it is necessary to calculate the deviation sequences before the calculation of the GRC. The deviation sequences are calculated using Equation (3) and its results are expressed in Table 11.

### 3.6.5. Principal Component Analysis

Pearson and Hotelling explain the structure of variance-covariance by way of the linear combinations of each quality characteristic initially developed PCA. It is lined up in descending order concerning variance, and therefore, the first principal component accounts for the most variance in the

data. The matrix consists of Eigenvalues, Eigenvectors, and quality characteristic contributions [48–50]. The principal component with the highest Eigenvalues is chosen to replace the original responses for further analysis. In this case, the highest Eigenvalues were obtained in the UTS first principal component as shown in Table 12. Then, the contributions of each quality characteristic for the first principal components are shown in Tables 13 and 14.

**Table 12.** Eigenvalues and explained variation for ultimate tensile strength and hardness.

| Principal Component | Eigenvalues | Explained Variation (%) |
|---|---|---|
| UTS | 1.9839 | 99.2 |
| HR | 0.0161 | 0.8 |

**Table 13.** The Eigenvectors for the principal component of ultimate tensile strength and hardness.

| Quality Characteristic | Eigenvector | |
|---|---|---|
| | 1st Principal | 2nd Principal |
| UTS | 0.707 | 0.707 |
| HR | 0.707 | −0.707 |

**Table 14.** Quality characteristic contribution of ultimate tensile strength and hardness.

| | |
|---|---|
| UTS | 0.4999 |
| HR | 0.4999 |

### 3.6.6. Calculation of Grey Relational Grades

Grey relational grade represents the level of correlation between the reference sequence and comparability sequence. Grey relational grade is a weighted average of the Grey relational coefficients of multi-objective [47]. It is determined using Equation (5) [38].

$$\gamma i(x_{0^*}, x_{1^*}) = \frac{1}{n} \sum_{i=1}^{n} wi\xi(x_{0^*}(k), x_{i^*}(k)) \tag{5}$$

where $\gamma i(x_{0^*}, x_{1^*})$ is the GRG for the *i*-th experiment, *wi* is the weighting value of the *i*-th performance characteristic, and n is the number of performance characteristics. The results of GRG with its ranks are depicted in Table 15.

**Table 15.** Results of Grey relational coefficient and Grey relational grade with its rank.

| Step 3: Grey Relational Coefficient | | | Step 4: Grey Relational Grade and It Is Rank | |
|---|---|---|---|---|
| No | UTS (MPa) | HR (HRA) | GRG | Rank |
| 1 | 0.542 | 0.427 | 0.485 | 7 |
| 2 | 0.644 | 0.618 | 0.631 | 5 |
| 3 | 0.770 | 0.713 | 0.741 | 4 |
| 4 | 0.395 | 0.361 | 0.378 | 8 |
| 5 | 0.551 | 0.542 | 0.546 | 6 |
| 6 | 1.000 | 1.000 | 1.000 | 1 |
| 7 | 0.333 | 0.333 | 0.333 | 9 |
| 8 | 0.841 | 0.796 | 0.819 | 3 |
| 9 | 0.949 | 0.845 | 0.897 | 2 |
| **Average GRG = 0.648** | | | | |

### 3.7. Analysis of Experimental Data

Experimentations are executed on a vertical CNC machine center according to the L9 orthogonal array arrangement. The hardness, tensile strength, and signal-to-noise-ratio results are depicted in Table 10.

Therefore, the Grey relational coefficient values are taken as ξ = 0.5.

Considering the highest GRG value for each parameter in Table 16, and the marked points in Figure 11, this indicates the optimal parameter setting of a rotational speed of 1400 rpm, and traverse speed of 37.5 mm/min, is an optimal parameter combination for the multiple performance characteristics. Based on the results presented in Table 16, rotational speed has the largest effect on the hardness and tensile strength of the welded joint.

**Table 16.** Response table of main effects for GRG.

| Level | Tool Profile (A) | Rotational Speed (B) | Traverse Speed (C) |
|-------|------------------|----------------------|--------------------|
| 1 | 0.6191 | 0.3988 | 0.7679 |
| 2 | 0.6416 | 0.6654 | 0.6355 |
| 3 | 0.6831 | 0.8795 | 0.5404 |
| Delta | 0.0640 | 0.4807 | 0.2275 |
| Rank | 3 | 1 | 2 |

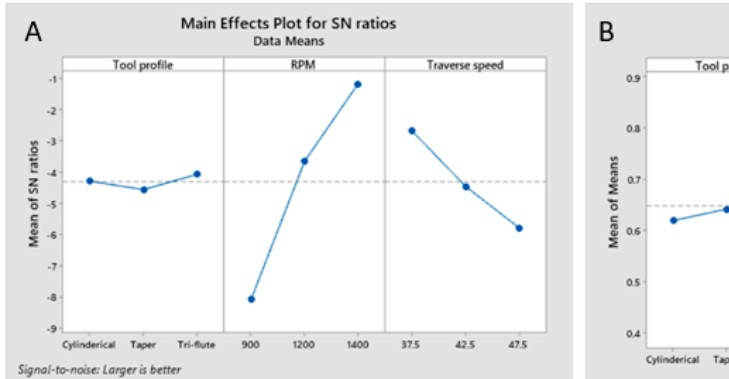 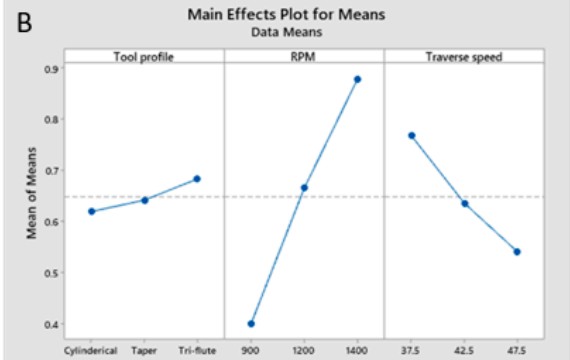

**Figure 11.** (**A**) Signal to noise ratio of Grey relational grade (GRG) (**B**) Main effects of Grey relational grade (GRG).

### 3.8. Optimal Combination of Each Factor Level

Quality of the process is measured in terms of particular target response to noise and signal factors as per Taguchi's experimental approach. Hence, quality enhancement attempted can be confirmed as effort to maximize the signal-to-noise (S/N)-ratio.

The aim of this research was to maximize the mechanical property of the weldment that can be categorized in 'maximum-the-better' quality criterion. Thus, S/N ratio (η) can be calculated as shown in Equation (1). Maximum values of S/N ratio were selected as the optimum level for each control factors. Based on experimental results of friction stir welding, the required responses were measured with appropriate measuring devices and summarized as shown in Table 7. Table 10 show measured responses and the S/N ratio value of each L9 orthogonal array that was calculated using Equation (6). Table 17 divulges optimum combination of control factors based on its highest average value of S/N ratios. In addition, main effect plot confirms it, as shown Figure 11. Hence, optimum combinations and levels have been found to be $A_3B_3C_1$; explicitly, a tool profile of tri-flute, rotational speed of 1400 rpm, and traverse speed of 37.5 mm/min have been found to be the optimal parameters.

**Table 17.** Optimum process parameters.

| Parameter | Level | Value |
|---|---|---|
| Tool profile | 3 | Tri-flute |
| Rotational Speed, rpm | 3 | 1400 |
| Traverse speed, mm/min | 1 | 37.5 |

To indentify which optimal process parameters are significant or not Anova analysis should be carried out. For this typical research, rotational speed of the tool and traverse speed become significant as shown in Table 18 and the model used is summarized and shown in Table 19.

**Table 18.** ANOVA results for Grey relational grade (GRG).

| Source | DF | Adj SS | Adj MS | F-Value | P-Value | Contribution | Remark |
|---|---|---|---|---|---|---|---|
| Tool profile | 2 | 0.006332 | 0.003166 | 27.78 | 0.035 | | Insignificant |
| RPM | 2 | 0.347993 | 0.173996 | 1526.44 | 0.001 | 80.33713576 | Significant |
| Traverse speed | 2 | 0.078329 | 0.039164 | 343.58 | 0.003 | 18.04209923 | Significant |
| Error | 2 | 0.000228 | 0.000114 | | | 1.62 | |
| Total | 8 | 0.432882 | | | | 100% | |
| $F_{0.01}(2,2) = 99$ | | | | | | | |

**Table 19.** Model summary.

| S | R-sq | R-sq(adj) | R-sq(pred) |
|---|---|---|---|
| 0.0106765 | 99.95% | 99.79% | 98.93% |

*3.9. Confirmation Experiment*

The confirmation experiment is a final step in the first iteration of the design of experiment process. The sample size of the confirmation experiment is larger than the sample size of any specific trial in the previous factorial experiment [51]. Therefore, this study conducted 10 experiments at the optimal condition ($A_3B_3C_1$) of the tool profile of the triflute threaded tool, a rotational speed of 1400 rpm, and a traverse speed of 37.5 mm/min. Summerized results of confirmation experimentsa are shown in Table 20.

**Table 20.** Results of the confirmation tests.

| Optimal Combination | The Response of Quality Characteristics | | | |
|---|---|---|---|---|
| $A_3B_3C_1$ | UTS | $S/N_{UTS}$ | HR | $S/N_{HR}$ |
| Replication 1 | 283 | 49.0357 | 72.0 | 37.1466 |
| Replication 2 | 284 | 49.0664 | 72.0 | 37.1466 |
| Replication 3 | 284 | 49.0664 | 71.6 | 37.0983 |
| Replication 4 | 284 | 49.0664 | 72.0 | 37.1466 |
| Replication 5 | 284 | 49.0664 | 72.0 | 37.1466 |
| Replication 6 | 284 | 49.0664 | 72.0 | 37.1466 |
| Replication 7 | 284 | 49.0664 | 72.0 | 37.1466 |
| Replication 8 | 284 | 49.0664 | 72.0 | 37.1466 |
| Replication 9 | 284 | 49.0664 | 72.0 | 37.1466 |
| Replication 10 | 283 | 49.0357 | 72.0 | 37.1466 |
| Mean of GRG for confirmation test = 0.933 | | | | |

A 99% confidence interval for the predicted mean of Grey relational grade (μGRG) on a confirmation test was calculated using the below Equations [47].

$$\mu A1B2 = \acute{I}GRG + (A1 - \acute{I}GRG) + (B2 - \acute{I}GRG) = A1 + B2 - \acute{I}GRG \tag{6}$$

where ÍGRG is the overall mean of Grey relational grade = 0.648, ÍGRG is equal to the overall mean of Grey relational grade = 0.648. $A_1$ and $B_2$ are the mean values of Grey relational grade with parameters at optimum levels.

$$\mu B_2 C_1 D_3 F_2 = 0.8795 + 0.7679 - 0.648 = 0.99994$$

The predicted mean of the Grey relational grade in the confirmation test is estimated by the following Equation: confidence interval for the predicted mean on a confirmation run is calculated using the below Equation.

$$CI = \mu \pm \sqrt{F\alpha; (1; fe) * Ve(\frac{1}{n_{eff}} + \frac{1}{r})} \tag{7}$$

where $F\alpha; (1, fe) = F_{0.01}; (2,2) = 99$

$\alpha$ = Risk = 0.01.
$fe$ = Error degree of freedom = 2.
$Ve$ = Error adjusted mean square = 0.000114.
$n_{eff}$ = Effective number of replications.
$R$ = Number of replications for confirmation experiment = 10.

In addition, the effective number of replications ($n_{eff}$) is calculated by:

$$n_{eff} = \frac{Tn}{1 + Ts} = \frac{9}{1 + 4} = 1.8 \tag{8}$$

where $n_{eff}$ = is expressed in mathematical.

$T_n$ = Total number of experiments = 9.
$T_s$ = the sum of the total degree of freedom of significant factors.

Therefore,

$$CI = 1.0307 \pm \sqrt{99 * 0.000114(\frac{1}{1.8} + \frac{1}{10})} = 0.0860 \tag{9}$$

The 99% confidence interval of the predicted optimal Grey relational grade is:

$$(\mu - CI) < \mu < (\mu + CI)$$

$$(0.99994 - 0.0860) < 0.99994 < (0.99994 + 0.0860) \tag{10}$$

$$0.91394 < 0.99994 < 1.08594$$

At 99% of the confidence interval, the predicted GRG at optimum condition is between 0.91394 and 1.08594. If the predicted and observed GRG values of the multiple performance parameters are close to each other, the effectiveness of the optimal condition can be ensured. To the test, the predicted results confirmation experiments were conducted ten times at the optimum condition. The Grey relational grade for the experiment is 0.933, which is in the range of the 99% confidence interval and achieved hardness and tensile strength of 283.9 MPa and 71.96 HR, respectively. Hence, the results of the confirmatory experiment tests show that the experiment was safest.

## 4. Influences of Significant Factors on Tensile and Hardness Strengths

In order to investigate the influence and effects of identified significant factors, experiments have been executed using optimal control factors. The two-process parameters found to have the greatest influences on weldment were rotational and traverse speeds. Heat contribution is directly correlated to rotational speed: augmented rotational speed upsurges heat input. Slow traverse speed results in a

higher heat input rate and a corresponding increase in weld temperature; hence, good weldment and enhanced mechanical properties (tensile and hardness) have been obtained.

The experimental investigation was conducted at the optimum process parameters, keeping optimum factors at their optimum level and altering significant factors accordingly at given levels the influence of these factors, as explained below.

### 4.1. Influence of Rotational Speed of on Tensile and Hardness Strengths

According to Kulekci, MK et al. [52] and G. Buffa et al. [53], tool rotational speed regulates the volume of plasticized substantial as well as material transportation. The amount of material becomes softer, more flexible, and material transportation from advancing side to retreading side grows as tool rotational speed surges. Therefore, the amount of material sited in advancing lateral grants the foremost share of the weld region. Experimental results (Figure 12), for this typical study, show that the 1400 rpm of rotation of the tool head gives better tensile strength. A friction stir welded joint shows determined strength when tool rotational speed offers suitable stirring of the plasticized substantial and good merging of the same at the retreading lateral. Meanwhile, smaller rotational speed exhibits smaller tensile strength. This is due to insufficient heat flux generation for a better weldment. At the same time, when rotational speed rises the microhardness of the weld zone approaches the microhardness of the recrystallized dominant structure of the weld zone reliant upon material location. As tool rotational speed reaches 1400 rpm, the weld zone hardness exhibits the maximum hardness number (Figure 13).

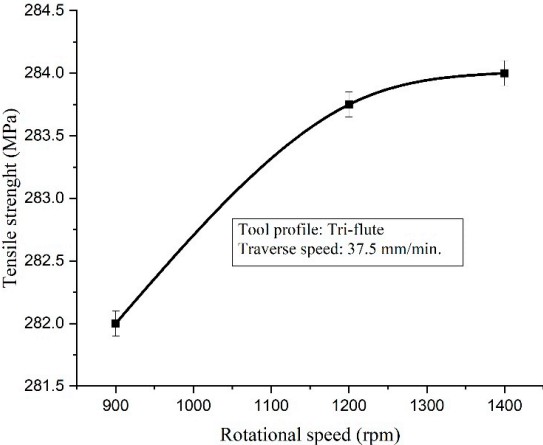

**Figure 12.** Influence of rotational speed on tensile strength.

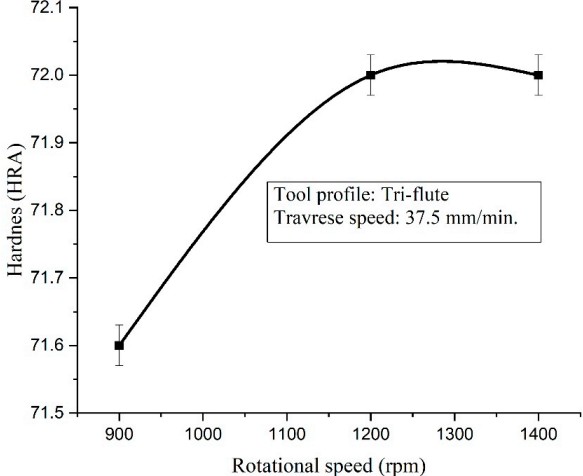

**Figure 13.** Influence of rotational speed on hardness strength.

### 4.2. Influence of Traverse Speed of on Tensile and Hardness Strengths

To explain the influence of the significant factors identified, traverse speed, on tensile and hardness strengths, successive experimental trials have been executed against its level. Keeping optimal parameters unchanged, tool geometry-cylindrical, rotational speed—1400 rpm, altering only traverse speed at 37.5 mm/min, 42.5 mm/min and 47.5 mm/min. The maximum tensile and hardness strengths were observed at 37.5 mm/min. are shown in Figures 14 and 15. This is because, according to investigations, the tool feed rate regulates the heat flux conferred to the joint during the process [53]. According to Luis Trueba Jr et al. [54] remarkably, situations that would have led to hot weld conditions (high rotational speed and low traverse speed) is one confirmation. This can be explained further by recalling that one of the purposes of the shoulders is to constrain plasticized weld metal and with increased traverse rates, the ability of the shoulders to constrain weld metal is reduced.

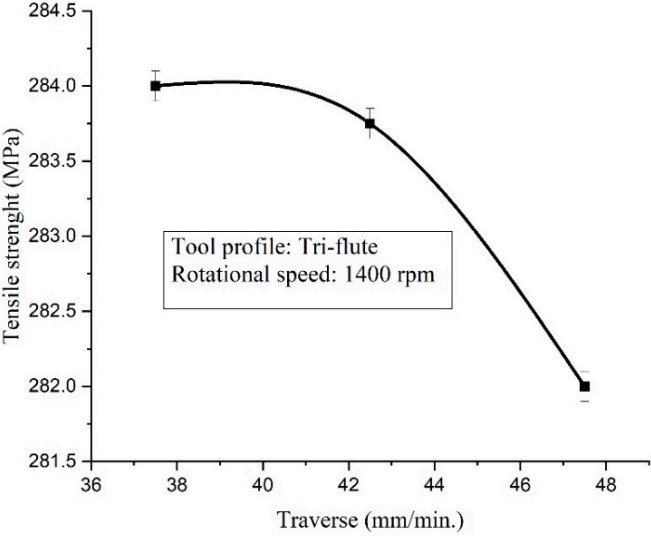

**Figure 14.** Influence of traverse speed on tensile strength.

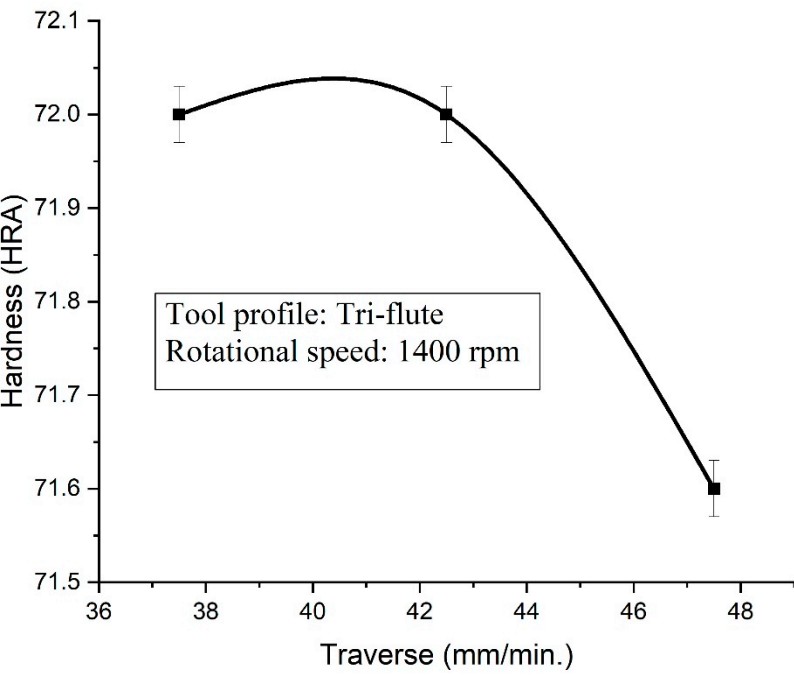

**Figure 15.** Influence of traverse speed on hardness strength.

## 5. Conclusions

In this study, a combination of Taguchi based Grey relational analysis method was implemented to come up with the optimal process parameters for FSW. Analyzing the effect of combined factors on the mechanical strengths, the following conclusions are drawn.

i.  From the experimental outcome and analyses of variance, one can conclude that the rotational speed and traverse speeds are significant parameters:

   a.  Significance shows a small change caused by the amount of this parameter will result in a diminished mechanical property of targeted quality criteria.
   b.  Altering the value of significant controllable factors will influence the formation of defects.

ii.  The highest hardness of 71.6 HR and tensile strength of 283 MPa was achieved at a parameter setting of the rotational speed of 1400 rpm, traverse speed of 37.5 mm/min, and tool shape of taper threaded pin. Similarly, the lowest hardness and tensile strength of 54.23 HR and 217 MPa respectively, were observed at a rotational speed of 900 rpm, traverse speed of 47.5 mm/min, and tri-flute threaded tools and, flash defect is found at the stir zone.

iii.  The rotational speed and traverse speed are sources of welding temperature. If the rotational speed increased, the welding temperature also increased and gets a maximum hardness and tensile strength. Traverse speed is indirectly proportional to the rotational speed, and welding temperature. In addition, the maximum temperature was obtained at a tapered tool pin profile of 416 °C, which is (36.19%) less than the temperature of (652 °C) of the liquid of the base material.

iv.  The advancing side gave higher temperatures than the retreating side, with the increment of rotational speed, and the temperature difference between the advancing and retreating side at the weld center varied from 11 °C to 23 °C.

v.  Based on the analysis of variance results, rpm has a greater effect, with an 80.33% contribution and the traverse speed effect has an 18.042% contribution.

vi.  For this material, a combination parameter of the tapered threaded tool with a rotation speed of 1400 rpm and a traverse speed of 37.5 mm/min imparts a sound weld.

**Author Contributions:** E.M. was in charge of the whole trial and wrote the manuscripts. R.A.-S. completes the whole analysis and assisted in modifying the structure and content of the paper. A.A. revised the final manuscript. All authors have read and agreed to the published version of the manuscript.

**Funding:** The authors would like to extend their acknowledgements to the School of Research and Graduate Studies, Bahir Dar Institute of Technology for their financial support. This reseach was funded with the grant number of BDU/BiT/SRGS/3320/2011.

**Acknowledgments:** The authors would like to acknowledge the School of Research and Graduate Studies, Bahir Dar Institute of Technology for their financial support.

**Conflicts of Interest:** The authors declare that they have no conflicting personal and financial interests.

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
