# Peer review of "Experimental Investigation of Friction Stir Welding on 6061-T6 Aluminum Alloy using Taguchi-Based GRA"

_metals, doi:10.3390/met10111480_

Round 1
Reviewer 1 Report
1. The problem presented in the work is interesting but paper is prepared without required explaination in some parts of the research, so it is difficult to asses submited results. Because this study is concentrated on identification of the optimum welding parameters for 6061 AA through Grey relational analysis and Taguchi method, the explaination concerning using them as statistical analysis exactly for FSW method should be presented and added to sub chapters : 2.3.1 2.3.2. , 2.3.3, 2.3.4., 2.3.5, 2.3.6. and 4.6.
2. All titles of Tables and all captions of figures do not reflect the contence of them.
e.g. Table 6 : Experimantal result , but of what ????????
e.g. Fig.5 Before and after the tensile test, but what ????
e.g. Fig. 7. Relationship b/n RPM and HR but kind of the process shoud be added and kind of material......
e.g. Fig.9 Liquid penetrant test , but it is not clear what is visiable on 3 parts of the picture. Explaination shoud be added as: a)..., b).......c)......
3. It was presented that severe plastic deformation exists in the FSW weld and it is caused by the high stirring process. To say in such a way, the SPD effect shoud be presented as characteristic microstructure of FSW joint zone ( cross sections ).
4. For better clarification of the problem of optimization of the FSW process parameters making use of the combination of Grey relation analysis method and L9 orthogonal array additional information should be included in ths work.
5. The effect of temperature ( as result of generated heat by friction) should be discussed taking into account the thickness of the sheet. Using samples of 5 mm in thickness , it is different case then for a.g. 0,5 mm.
6. In which way the identification of defects in FSW joints was realized ? ( e.g. non-destructive tests? ) ( Table 8).
Author Response
Dear reviewer 1,
Thank you very much for your time and valuable comments.
Attached, please find the responses.

Reviewer 2 Report
1 - Page 3, line 86
Which is the temper of the aluminium alloy?
2 – Page 3, line 92
Was the chemical composition determined by the authors?
3 – Page 3, line 96
Which is the size of the plates welded?
4 – Page 4, Fig. 3
Fig. 3 is not clear, it was better to do a scheme.
5 – Page 4, line 114
Figs 4 and 5 are not mentioned in the text. No comment about the tools.
Which were the welding parameters used in the tests?
6 – Page 6, Fig. 6
Why axial load or tool plunge depth were not considered.
7 – Page 8, line 185
Why the welds present these properties? Because of defects? Because of mechanical properties. Where does the failure occur?
8 – Page 8, line 190
Where does the failure occur? Why measure the hardness in the nugget and not in the HAZ?
Is there any correlation between the hardness and the failure of the specimens in this case?
9 – Page 8, line 202
It is not understood why there is a crack, as it is not a fusion welding process. Unless it is a ditch on the surface, in which case the defect is poorly classified. In Fig. 9 it seems to be a ditch.
10 – Page 9, table 7
Put units at welding speed.
11 – Page 16, line 294
Did you check the internal defects?
Author Response
Dear reviewer 2,
Thank you very much for your time and valuable comments.
Attached, please find the responses.

Reviewer 3 Report
All the comments are reported in the attached pdf file.

Author Response
Dear reviewer 3,
Thank you very much for your time and valuable comments.
Attached, please find the responses.
